# Multiple Resistive Switching Mechanisms in Graphene Oxide-Based Resistive Memory Devices

**DOI:** 10.3390/nano12203626

**Published:** 2022-10-16

**Authors:** Sergei Koveshnikov, Oleg Kononenko, Oleg Soltanovich, Olesya Kapitanova, Maxim Knyazev, Vladimir Volkov, Eugene Yakimov

**Affiliations:** 1Institute of Microelectronics Technology, Russian Academy of Sciences, Chernogolovka 142432, Russia; 2Department of Chemistry, Moscow State University, Moscow 119991, Russia

**Keywords:** graphene oxide, resistive switching, operative mechanism, electron beam-induced current

## Abstract

Among the different graphene derivatives, graphene oxide is the most intensively studied material as it exhibits reliable and repeatable resistive switching. The operative mechanisms that are responsible for resistive switching are being intensively investigated, and three models explaining the change in the resistive states have been developed. These models are grounded in the metallic-like filamentary conduction, contact resistance modification and the oxidation of/reduction in the graphene oxide bulk. In this work, using Al/GO/n-Si structures, we demonstrate that all three of these operative mechanisms can simultaneously participate in the resistive switching of graphene oxide. Multiple point-like conduction channels in the graphene oxide films were detected by the electron beam-induced current (EBIC) technique. At the same time, large areas with increased conductivity were also revealed by EBIC. An analysis of these areas by Raman spectroscopy indicates the change in the graphene oxide bulk’s resistive properties. The EBIC data along with the measurements of the capacitance–voltage characteristics provided strong evidence of the involvement of an aluminum/graphene oxide interface in the switching processes. In addition, by using Al/GO/n-Si structures, we were able to identify unique local properties of the formed conductive channels, namely the change of the charge state of a conductive channel due to the creation of negatively charged traps and/or an increase in the GO work function.

## 1. Introduction

The research and development of materials, devices and processes that are related to non-volatile resistive memory is of great interest due to the prospects of the important practical applications including large-volume memory arrays [1], systems-on-chip [2] and neuromorphic computing systems [3]. Two reliable resistive states, high-resistance (HRS) ones and low-resistance (LRS) ones, along with a high switching speed, low power and a high number of cycles, make the elements of resistive memory attractive for their implementation in digital electronics, while them achieving multiple intermediate resistive states allows for the development of artificial neural networks, in which resistive memory elements can act as artificial synapses [4]. These possibilities stimulate the intensive scientific research of the widely used dielectrics, such as metal oxides [1] and new materials, among which the various graphene derivatives, e.g., graphene oxide (GO), graphene nitride and graphene fluoride, are of rapidly growing interest [5,6,7,8,9,10]. Graphene oxide is the most intensively studied material as it exhibits reliable and repeatable resistive switching [8,11,12]. However, unlike the case of the metal oxides, for which the switching mechanisms have been thoroughly investigated and strong models have been developed [13,14], the operative mechanisms of resistive switching in graphene oxides have not yet been fully elucidated due mainly to a complex structure of the GO film [15,16]. To determine the switching mechanisms, the research is being focused on the evaluation of the fabrication/deposition methods, electrode materials, control of the degree of the reduction in/oxidation of the graphene oxide/graphene structures, the determination of its structural properties and its chemical composition [17]. The three major models that have been developed so far to explain the changes in the resistive state of graphene oxide under an electric field are grounded in the concepts of (1) metallic-like filamentary conduction [5,6,11], (2) contact resistance modification due to ion drift [7] and (3) the oxidation of/reduction in the graphene oxide bulk [8]. Filament-based switching can occur due to the drift of the oxygen vacancies under an electric field, thus leading to formation of conductive channels between the graphene oxide flakes or at their interfaces [18]. The drift of the oxygen ions near the GO/metal interface, e.g., near the aluminum electrode, can lead to a change in its resistance due to the formation of a local conductive filament near the interface with the volume of graphene oxide. The change of the properties of the bulk graphene oxide is assumed to be due to the transition of the sp3 domains with dielectric properties to a conducting state that is associated with two sp2 bonds [8].

In our previous work [19], the HRS-to-LRS transition in the Al/GO/n-Si structures were shown to occur under both positive and negative voltage sweeps, indicating the formation of multiple conductive filaments between the GO layers. Stable non-volatile switching was explained by capturing and emitting the electrons by deep-level traps that exist in the GO film, and it was concluded that at least two resistive switching mechanisms work simultaneously.

In this work, we used Al/GO/n-Si metal–insulator–semiconductor (MIS) structures to investigate the resistive switching mechanisms and determine the locations of the conductive channels in the graphene oxide films by the electron beam-induced current (EBIC) technique [20,21]. To the best of our knowledge, the EBIC method was used for the first time to reveal conductive channels that are formed in the graphene oxide due to resistive switching. Although several switching mechanisms including metallic-like filamentary conduction, contact resistance modification and the oxidation of/reduction in the graphene oxide bulk have been previously discussed, our work demonstrates that all three of the mechanisms govern the resistive switching in graphene oxide, simultaneously. In addition, by using the Al/GO/n-Si structures, we were able to identify the unique local properties of the conductive channels. These properties include the change in the charge state of a conductive channel, and this change is caused by either the creation of negatively charged traps or an increase in the GO work function.

## 2. Materials and Methods

The graphene oxide that was used in this work was synthesized by the oxidation of graphite using the modified Hummers’ method [22,23], and it was deposited on the n-type Si substrates by spin-coating 5 mg/mL of the GO-based sol at the rotation speed of 3000 rpm. The GO films that we used in the present study were the same as those which we investigated in our previous work [19], where the structural and electronic properties were characterized in detail by scanning electron microscopy, atomic force microscopy as well as by infrared, Raman, X-ray absorption and photoelectron spectroscopies. Here, we briefly note that the GO film under the study consists of flakes with a size that is between 2 and 10 µm. Based on the XPS and Raman spectroscopy measurements, the oxygen content in the GO films was estimated to be 35 +/− 5%. An analysis of the Raman and photoelectron spectra showed that the GO films consist of sp2 graphene domains in an oxygen-enriched sp3 matrix.

The investigations of the resistive switching characteristics and operative switching mechanisms in the GO films were carried out on the MIS structures that were made on phosphorous-doped Si wafer pieces with the GO film thickness of 40 nm. The top electrodes of circular shape and 1 mm in diameter were made by the sputtering of Al or Pt; the ohmic contact to the bottom Si electrode was formed by scratching the InGa paste. The use of the MIS structures with the bottom electrode of the n-type silicon allows us to determine the charges in the graphene oxide, evaluate its dielectric permittivity and provide the injection of holes under the negative bias that was applied to the top electrode. The use of aluminum as a top metal electrode allows the experimental verification of one of the switching mechanisms that is listed above, whereas Al fusibility can be employed to visualize the conductive channels under a high current flow. The use of the MIS structures also makes it possible to apply the method of electron beam-induced current to detect the conductive channels that are formed during the resistive switching processes. The GO film switching characteristics were studied by the DC current–voltage measurements using the Keithley 2450 Source Meter. To evaluate the presence of the charge traps in the GO film, the capacitance–voltage (C–V) characteristics were measured using the Keithley 4200A-SCS parametric analyzer at 100 kHz. The EBIC study was carried out using the scanning electron microscope (SEM) Jeol JSM-840 and using the Keithly 428 current amplifier at room temperature with a beam energy that was in the range from 8 keV to 30 keV and a beam current that was lower than 10^−11^ A to minimize the irradiation dose. To monitor the e-beam irradiation effect during the EBIC mapping, the switching characteristics were measured on more than 20 MOS structures before and after their irradiation with an electron beam. After the experiments on resistive switching and the EBIC characterization, the aluminum electrodes were chemically removed, and the underlying GO film was characterized using the SENTERRA Bruker Raman microscope at the laser wavelength of 532 nm. A schematic presentation of the MIS structure and the experimental setup for the current–voltage and EBIC measurements is given in Figure 1a. The device was placed into the SEM chamber, where an electron beam was turned off during the measurements of current–voltage characteristics. An electron beam generates electron–hole pairs at different depth depending on the beam energy. As it is discussed below, the electrons and holes are separated by an internal electric field; the generated holes are driven towards the Al electrode and contribute to the collected current, the value of which determines the brightness of an EBIC image. Figure 1b presents the equivalent circuits of the Al/GO/n-Si MIS structure under the *AC* and *DC* measurements, respectively. Note that the capacitance playing a major role during the C–V measurements at a frequency of 100 kHz had an infinite value during the *DC* voltage sweep. The presence and role of aluminum oxide are explained in Section 3.4.

## 3. Results

### 3.1. Resistive Switching and Visualization of Conductive Channels by EBIC

Figure 2 presents the current–voltage characteristics of the Al/GO/n-Si MIS structures before and after the resistive switching occurred. The current–voltage characteristics which were measured on a pristine device at a low negative bias show a slight increase in the current after the low-dose electron irradiation (blue and red curves in inset of Figure 2a). The EBIC measurements of the pristine device demonstrate a low induced current, which is comparable with the e-beam current across the entire device area (Figure 3a). Some slight variations of the induced current, which do not exceed 10^−10^ A, can be caused by a non-uniformity of the Al film thickness or they may reflect the difference in the conductivity of the GO flakes. An increase in a negative voltage during the subsequent I–V measurements up to 10 V resulted in a sharp increase in the current at about −5 V (Figure 2a, green curve), thus indicating the transition from a high-resistance state (HRS) to a low-resistance state (LRS). As a result of this transition, a single bright spot appeared in the EBIC image (marked by yellow arrow in Figure 3b,c). A further increase in the negative bias up to −15 V caused the additional transitions to LRS as reflected by a step-like increase in the current (Figure 2a, cyan curve). Following the I-V measurements, the EBIC measurements reveal an appearance of multiple conductive channels that are seen as bright spots (Figure 3d). It should be emphasized that the values of the induced current measured at the bright spots exceed the value of the beam current by two-to--three orders-of-magnitude. A number of these bright spots were notably increased (Figure 3e) after sweeping the negative voltage up to −25 V (Figure 2b, red curve). When the voltage polarity during the subsequent I-V measurements was changed to a positive one, a sharp drop in the current occurred at about 12 V (Figure 2b, blue curve). Such a notable decrease in a current flowing through the graphene oxide film indicates a reverse transition from LRS to HRS. As a result of this transition, many of the conductive channels disappeared from the EBIC image (compare Figure 3e,f). At the same time, several new bright spots (marked with white arrows in Figure 3f) appeared despite a significant decrease in the current during the LRS-to-HRS transition (Figure 2b, blue curve). Thus, the formation and elimination of multiple conductive channels in the graphene oxide film during resistive switching can be visualized by the EBIC measurements.

### 3.2. Mechanism of EBIC Image Formation

As the resistive switching in the GO-based MIS structures has not been previously studied by the EBIC method, the mechanism of the conductive channel image formation should first be explained. The determination of this mechanism will be helpful to obtain a better understanding of the operative mechanisms that are responsible for resistive switching. The measurements of the EBIC current that is responsible for the appearance of the single bright spot (Figure 3b) show that its values exceeds the beam current by more than two-orders-of-magnitude. The high values of the induced current can be explained under an assumption that the main contribution to the induced current is provided by the excess minority carriers (i.e., holes) that were generated in n-Si. The contribution to the total induced current by the other components including the current which was generated inside the GO film and related to the hot-carrier injection from Si and/or Al cannot noticeably exceed the beam current due to a small thickness of the GO and Al layers. At the same time, a number of the carriers that are generated in Si is high enough to provide the measured current. It is important to note that during EBIC measurements, the bright spots were detected even when no external bias was applied to the structure. This implies that a local rectifying contact separating the electrons and holes must have been formed at the conductive channel/Si interface during the resistive switching. In this case, the conductive channels in the GO film can electrically connect the Si substrate with the Al contact by a flux of holes. The contribution of the holes to the induced current which was responsible for appearance of bright spots was proven via an experimental determination of a sign of the induced current, thus providing strong evidence of the formation of local rectifying barriers near the conductive channel/Si interfaces. To produce the rectifying contact which allows the holes to drift to the top electrode, a conductive channel near its interface with Si should be charged negatively. This can be accomplished due to the creation of the negatively charged traps and/or an increase in the GO work function. It should be noted that the EBIC demonstrated a different stability of the conductive channels under the reverse bias sweep during the LRS-to-HRS transition (Figure 2b, blue curve). As it can be clearly seen from Figure 3e,f, some of the conductive channels disappeared, while the others were still detected by the EBIC. This can be due to the different origins of the conductive channels and/or the different mechanisms of their formation.

### 3.3. Impact of High Current on Formation of Conductive Channels during Resistive Switching

As shown above, a progressively increased *DC* voltage results in an increase in the number of the bright spots that are revealed by the EBIC (Figure 3). The comparison of the EBIC images with the images that were obtained in the secondary electron (SE) mode shows that the bright spots are not associated with the surface defects and/or surface morphology variation. This implies that the bright spots are associated with the conductive paths that are formed in the GO. An additional piece of evidence of such an association is demonstrated by the EBIC image that is presented in Figure 3f, which shows the disappearance of some of the bright spots after the resistive switching from LRS to HRS under a reverse voltage polarity.

Figure 4a presents an EBIC image that was obtained using another pristine Al/GO/n-Si MIS device after the HRS-to-LRS transition at a high compliance current of 100 mA. Such a high current flowing through the graphene oxide results in a dramatic increase in a number of conductive channels and a significant enhancement of the EBIC intensity. Strong effect of a high current is also revealed by SEM in the SE mode. In this case, the transition from HRS to LRS is accompanied by a huge amount of Joule heating that leads to the local changes of the Al surface morphology (Figure 4b). By comparing the SE and EBIC images, one can observe several interesting features. First, some of the spots that are visible in the SE mode coincide with the positions of the conductive channels which are revealed by the EBIC. Second, the larger the local modification of the aluminum surface was, then the less the intensity of the EBIC signal at the same location was. This is due, most likely, to the fact that the structural modification of aluminum can disrupt the current path during the EBIC measurements. At the same time, the EBIC reveals a significantly larger number of the conductive channels than that which can be seen in the SE mode. Some of these channels are seen as individual bright spots with different intensities of an induced current, while the others seem to be agglomerated.

It is important to note that the transition to LRS at a high maximum current of 100 mA results in the appearance of large areas with an increased and rather uniform intensity of the induced current (Figure 5a). To find out the origin of these high-conductivity areas, the Al layer was chemically removed, and the exposed graphene oxide was analyzed by Raman spectroscopy within the selected area that is shown in Figure 5a. The Raman G-peak intensity map of this selected area is shown in Figure 5b. The intensities of both the G and D lines (Figure 5c, black curve) are quite similar across the measured area, thereby indicating a rather uniform thickness of the GO layer with the exception of a single spot where multiple GO flakes can be stacked in a similar manner to that which is shown in Figure 5d. Therefore, the large GO areas with an increased and uniform conductivity are not due to the GO thickness variations.

### 3.4. C–V Characteristics of Al/GO/n-Si MOS Structures

The pristine GO-based structures demonstrate an effective modulation of the MIS capacitance under an applied voltage. Indeed, the capacitance change from accumulation to depletion and vice versa occurs within a voltage range that is from +1 V to −1 V, thus indicating the good quality of the GO/Si interface (Figure 6a, blue curve). The C–V curves that were obtained under the voltage sweep from a positive bias of 1 V up to various negative voltages are shown in Figure 6a. The results of the measurements exhibit two essential features. First, a negative shift of the C–V curve along the voltage axis indicates the presence of a positive charge in the pristine GO film. The voltage shift of the C-V curve increases with an increase in the voltage sweep range that drives the MOS structure in depletion. At the same time, the C–V curves exhibit a clock-wise hysteresis, which also increases with an increasing the voltage sweep range. Some of the hole traps that are responsible for the C–V hysteresis are very shallow, and they released the charge when the bias was off, while the other traps were deep enough to hold the charge at room temperature for an extensive period of time. Second, the accumulation capacitance progressively increases with an increase in the voltage sweep range that drives the MOS structure in depletion. The dependencies of the C–V curve shift along the voltage axis, and the changes in the accumulation capacitance on the sweeping voltage range in depletion are presented in Figure 6b. It should be noted that the MIS structures with the Pt electrode revealed the same clock-wise hysteresis of the C–V curves, whereas no increase in the accumulation capacitance was observed with an increase in the voltage sweep range, the value of which depleted. The effect of the increase in the accumulation capacitance was observed on 20 MIS devices with different electrodes. The cumulative plot that shows the change of the accumulation capacitance of the MIS structures with the Al and Pt gates is presented in Figure 6c.

## 4. Discussion

The results of the current–voltage and EBIC measurements that are presented in Figure 2 and Figure 3, respectively, indicate that the HRS-to-LRS transition under the voltage sweep results in the formation of multiple conductive paths in the graphene oxide. Due to a complex structure of the graphene oxide film, both the number and locations of the point-like conductive channels across the film area are quite random and vary from device to device. Some of these conductive channels disappear under the voltage sweep of the opposite polarity, while several new channels are formed. This feature may indicate either a different origin of the formed conductive channels, or a different mechanism that governs the resistive switching. Some of the conductive channels can be associated with metallic-like conductive filaments, while the others—with the presence of sp2 graphene domains in an oxygen-enriched sp3 matrix of a pristine film or with a local thinning of the insulating GO layer, thereby resulting in a correspondent increase in an electric field.

Based on the analysis that is presented in Section 3.2, it was shown that the detection of the conductive channels in the Al/GO/n-Si MIS structures by EBIC was possible due to the formation of local rectifying contacts at the conductive channel/Si interfaces. A rectifying contact that is formed during resistive switching must over-compensate the impact of a positive charge (Figure 6a) in the bulk of the pristine GO film in order to separate the electrons and holes using a built-in internal electric field. This is a fundamentally important requirement for the drift of the holes towards the top electrode and for their contribution to the collected current that is responsible for detection of the conductive channels by EBIC. The evidence of the major contribution of the holes that are generated by an electron beam to the collected current was experimentally established (Section 3.2). Although the origin of such a rectifying barrier is unknown, it is reasonable to suggest that an applied electric field results in either creation of negatively charged traps or an increase in GO work function near the interface of a conductive channel with silicon. The formation of the rectifying contacts implies that all of the conductive channels that are revealed by the EBIC must be located in close vicinity to or directly at the GO/Si interface.

The transitions from HRS to LRS are seen when a negative voltage was applied to the top electrode (Figure 2). In this case, the multiple local conductive channels that were revealed by EBIC (Figure 3) could be formed due to the upward drift of the positively charged oxygen vacancies in the GO or caused by the drift of cations in the opposite direction. Along with an observed decrease in the current (Figure 2b), some of these conductive filaments were found to disappear under the applied voltage of an opposite polarity (Figure 3f), thus demonstrating a typical bipolar resistive switching behavior. These findings strongly support the model of metallic-like filamentary conduction [5,6,11]. Therefore, it is reasonable to assume that the physical mechanism that is responsible for the LRS-to-HRS transition is based on the destruction of the conductive filaments, consisting of either oxygen vacancies or cations that have accumulated near/at the GO/Si interface.

According to the data in [8], the conductive filaments consisting of the sp2 fragments in the sp3 domain can be formed under an electric field and provide stable conductive paths. The EBIC data that were obtained using the devices after the resistive switching at a high compliance current of 100 mA indicate that some of the conductive channels occupy a large area that is comparable to the size of the GO flakes (Figure 5a,d). Although a Raman investigation of these areas (Figure 5b,c) did not reveal direct evidence of the sp3-to-sp2 transition in the GO bulk, the data clearly indicate that the large GO areas with an increased and uniform conductivity were not related to the GO thickness variations. The experimental verification of the sp3-to-sp2 transitions in the GO bulk under an electric field should be continued on the structures with semi-transparent metal electrodes to avoid the GO film oxidation during the procedure of the metal removal.

The role of aluminum oxide in the resistive switching of the Al/GO structures was discussed in [7], where it was demonstrated that the surface of the GO film was reduced, while the aluminum at the interface with GO was oxidized. The presence of a thin aluminum oxide layer in the Al/GO/n-Si MIS structures that were used in the present work was demonstrated by results of the capacitance–voltage measurements (Figure 6). In particular, the accumulation capacitance was found to notably increase after increasing the negative voltage that was applied to the top electrode to at least 4 V. (Figure 6a). This can be explained by a series connection of graphene oxide with aluminum oxide (see *AC* equivalent circuit in Figure 1b). A negative bias that is applied to the Al electrode initiates the upward drift of the oxygen vacancies which create conductive paths in a thin aluminum oxide layer that, in turn, results in the electrical shortening of the aluminum oxide. As a consequence, the accumulation capacitance is determined by the thickness of the GO layer only, and hence, it increases. The values of both of the capacitances (the capacitance of graphene and aluminum oxides which are connected in series and the capacitance after the aluminum oxide shortening) are known from the C–V characteristics (Figure 6b). Therefore, the capacitance of the aluminum oxide layer and hence its thickness can be determined. Assuming that the aluminum oxide dielectric constant is in a range that spans from seven to nine, then the aluminum oxide thickness is estimated to be between 2.5 nm and 3.2 nm. It should be noted that the conductive channels in the aluminum oxide cannot be directly detected by the EBIC due to the GO layer separating them from the GO/Si interface. However, due to their contribution to the series resistance, the conductive channels in the aluminum oxide can affect the brightness of the EBIC images that are shown in Figure 3. Thus, the observed data strongly support the model of the contact resistance modification [7]. In this model, the resistive switching is attributed to the formation and rupture of the local filaments in a thin insulating layer at the interface with a top Al electrode.

To illustrate the formation and disappearance of the conductive channels in the Al/GO/n-Si MIS structure, Figure 7 presents the simple equivalent circuits of the device before and after the resistive switching. A symbol of a diode in Figure 7a illustrates a potential barrier that is formed at the GO/Si interface due to a positive charge in the pristine GO film. After the HRS-to-LRS transition (green dash-lined rectangle in Figure 7b), the initial resistance of the aluminum oxide layer R_AlO_ was strongly reduced to R*_AlO_ due to the formation of multiple metallic-like conductive filaments under relatively low voltages. Conductive filaments in the GO (R*_GO_) are formed at higher voltages, and they can be directly detected by the EBIC due to the formation of local rectifying barriers at the GO/Si interface. These local barriers over-compensate a barrier in the pristine GO film, as shown by the change in the diode polarity in Figure 7b. Some of the local conductive channels in the GO can be aligned with the conductive filaments in the aluminum oxide as denoted by a thicker line connecting the GO and the aluminum oxide layers. After the LRS-to-HRS transition (blue dash-lined rectangle in Figure 7b), some of the conductive channels in the GO disappear (Figure 3f) due to a full or partial elimination of the conductive filaments. The transition of some of the channels to HRS is accompanied by a polarity change in a rectifying barrier (denoted as 2*), while no barrier change occurs for the other channels (denoted as 1*). This difference illustrates the different mechanisms governing the change in the conductive channel resistance, i.e., reduction in the length of a conductive filament and/or sp2-to sp3-transition. The sub-circuit 3* illustrates the case when some channels remain unchanged during the LRS-to-HRS transition, while some of the new channels are formed as observed in the EBIC (Figure 3f).

The presence of the GO bulk traps for the holes was established by measuring the GO-based MIS capacitors (Figure 6). A notable shift of the C–V curve (Figure 6a) indicates the accumulation of a positive charge in the GO bulk traps. Although the specific location of these traps within the GO bulk is not known, they can participate in the resistive switching processes.

## 5. Summary

We have investigated the operative mechanisms that are responsible for the resistive switching in the graphene oxide-based MIS structures. To the best of our knowledge, the EBIC method was used for the first time to reveal the conductive channels that are formed in graphene oxide due to resistive switching. The employment of the EBIC method enabled us to detect multiple point-like conductive channels under applied voltages of both polarities. Some of these channels can be disrupted under the bias of a reversed polarity and that is typical for bipolar resistive switching, while some new conductive channels can be formed. In addition, by using the Al/GO/n-Si MIS structures, we were able to identify unique local properties of the conductive channels. These properties are associated with a change in the charge state of a conductive channel near/at the GO/Si interface. This change is caused by either the creation of negatively charged traps or an increase in the GO work function. By increasing the maximum current that is flowing through the MIS structure, the conductive channels can be visualized by local changes to the aluminum electrode morphology. At the same time, large areas with an increased conductivity are revealed by the EBIC, and they may indicate the change of the GO bulk resistive properties. The important role of aluminum oxide in resistive switching is revealed by the C–V characteristics. An unusual dependence of the accumulation capacitance of the Al/GO/n-Si MIS structure on the bias that is in depletion is explained by the presence of a thin aluminum oxide which is connected in a series with the graphene oxide. The drift of the oxygen vacancies that are accumulated in the top GO layer is believed to be responsible for the formation of conductive filaments in a thin aluminum oxide layer. The formation of the conductive channels results in an increase in the conductivity and electrical shortening of the aluminum oxide. The presence of traps for the holes in the GO bulk may affect the resistive switching processes by accumulating and releasing the positive charge.

## 6. Conclusions

In conclusion, at least two out of three major resistive switching mechanisms, namely metallic-like filamentary conduction and contact resistance modification due to the drift of the oxygen vacancies can operate simultaneously in the graphene oxide-based MIS structures. The third mechanism which is related to the oxidation of/reduction in the graphene oxide bulk due to the sp2–sp3 transitions cannot be excluded from consideration, but it needs additional experimental verification. The conductive channels in the GO can be revealed by the EBIC only when the local rectifying barriers are formed near/at the GO/Si interface. The creation of negatively charged traps or the change in the value of the GO work function can be responsible for the formation of such rectifying barriers.

## Figures and Tables

**Figure 1 nanomaterials-12-03626-f001:**
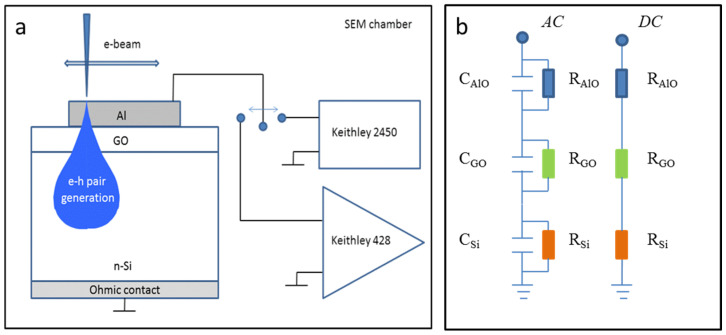
(**a**) Schematic presentation of the Al/GO/n-Si MIS structure and the experimental setup for current–voltage measurements using Keithley 2450 Source Meter and EBIC characterization using Keithley 428 Current Amplifier; (**b**) equivalent circuits of the Al/GO/n-Si MIS structure under *AC* and *DC* measurements, respectively.

**Figure 2 nanomaterials-12-03626-f002:**
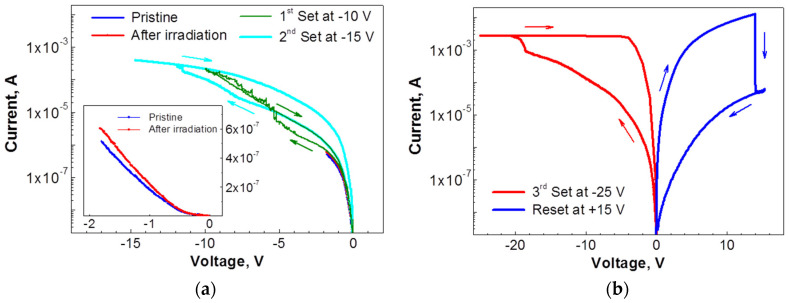
Current–voltage characteristics of Al/GO/n-Si MIS structures: (**a**) before and after e-beam irradiation of pristine device and after first and second Sets of negative voltage sweeps of up to −10 V and −15 V, respectively. Inset shows I–V characteristics of the pristine device and e-beam-irradiated device on linear scale; (**b**) after third Set voltage sweep of up to −25 V, and subsequent Reset using voltage sweep of up to +15 V. Arrows show direction of the voltage sweep.

**Figure 3 nanomaterials-12-03626-f003:**
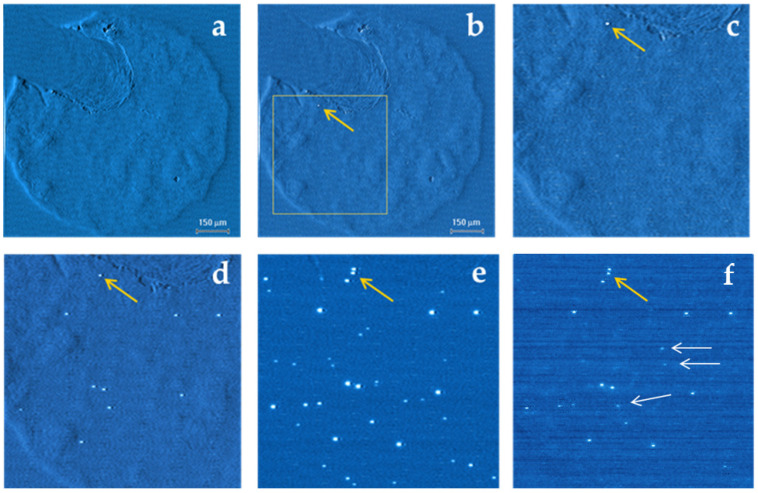
EBIC images obtained on Al/GO/n-Si MIS structures: pristine device (**a**); after Set voltage sweep of up to −10 V (**b**); images of fragment marked as yellow square after Set voltage sweeps of up to −10 V (**c**), −15 V (**d**), −25 V (**e**), and Reset voltage sweep of up to +15 V (**f**). Yellow arrows show location of the first-formed conductive channel; white arrows show new conductive channels formed during positive voltage sweep.

**Figure 4 nanomaterials-12-03626-f004:**
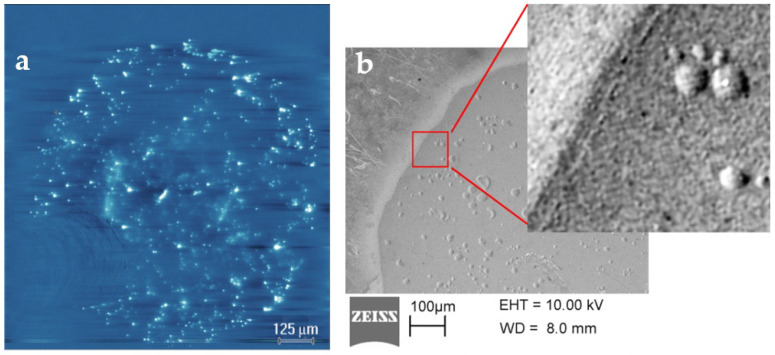
EBIC images obtained on Al/GO/n-Si MIS structure after HRS-to-LRS transition at a compliance current of 100 mA (**a**); SE images of selected device areas showing modification of the aluminum surface (**b**).

**Figure 5 nanomaterials-12-03626-f005:**
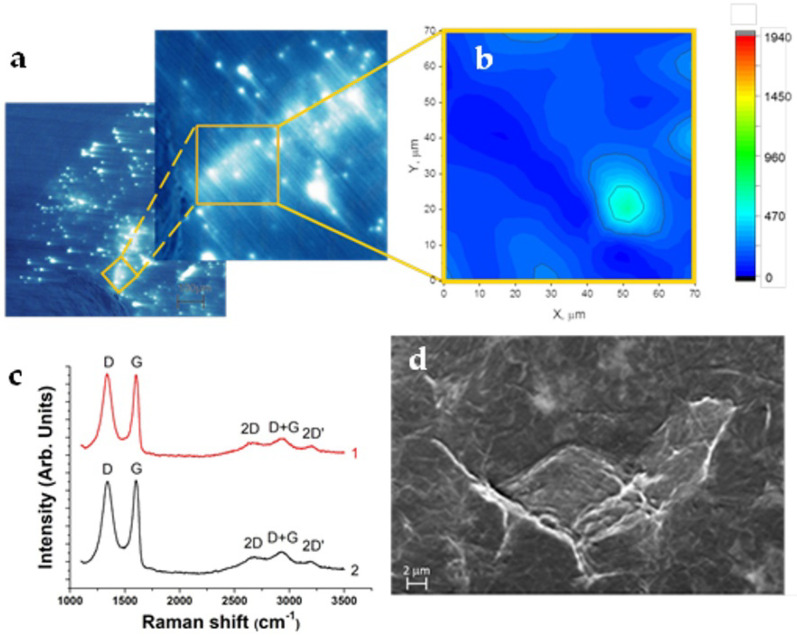
(**a**) EBIC image of a selected area showing higher and uniform intensity of the induced current; (**b**) Raman G-peak intensity map of the selected area marked in (**a**); (**c**) Raman spectra obtained on pristine GO film (red curve 1) and after resistive switching at maximum compliance current of 100 mA (black curve 2); (**d**) plain view SE image of pristine GO film.

**Figure 6 nanomaterials-12-03626-f006:**
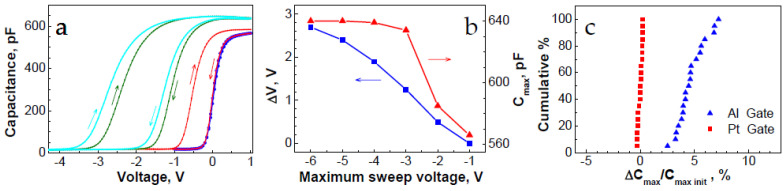
(**a**) Capacitance–voltage characteristics of Al/GO/n-Si MIS structures measured from accumulation at +1 V to depletion at negative bias up to −1 V (blue), −2 V (red), −5 V (green) and −6 V (cyan)Arrows show direction of the voltage sweep; (**b**) C–V curve shift along the voltage axis during sweep from accumulation to depletion (blue), and maximum capacitance (red) after sweep at various negative voltages as a function of maximum negative sweep voltage; (**c**) cumulative plot of the change of accumulation capacitance measured on MIS structures with Al and Pt gates.

**Figure 7 nanomaterials-12-03626-f007:**
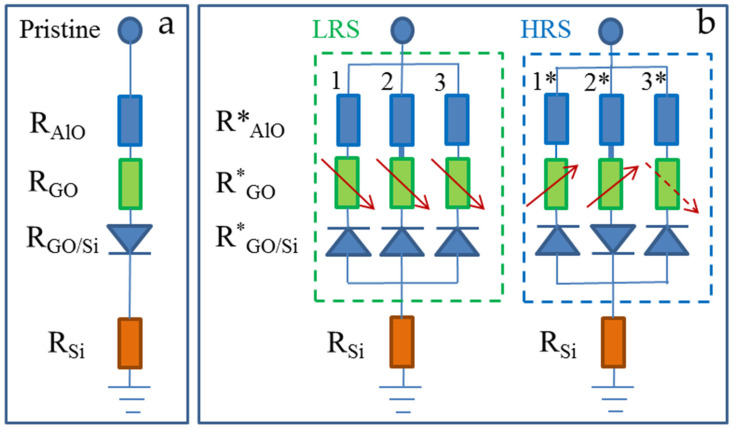
Equivalent circuits of the Al/GO/n-Si MIS structure: (**a**) pristine device; (**b**) illustration of conductive channels formed under *DC* voltage sweep resulting in HRS-to-LRS (green dash-lined rectangle) and LRS-to-HRS transition (blue dash-lined rectangle). Directions of arrows indicate changes in resistance of conductive channels. Symbols of diode illustrate the presence of rectifying contacts at the GO/Si interface due to either positive charge in pristine GO film or formation of conductive channels near/at the GO/Si interface. The meaning of asterisks is explained in the text.

## Data Availability

The data presented in this study are available on request from the corresponding author.

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
