# Peer review of "Multiple Resistive Switching Mechanisms in Graphene Oxide-Based Resistive Memory Devices"

_nanomaterials, 2022, doi:10.3390/nano12203626_

Round 1

Reviewer 1 Report

This manuscript summarizes EBIC experiments on graphene oxide films.  I think this manuscript needs to be improved so the reader can understand the experiments and results. 

1. The system as the authors describe it is not clear. The spin coated film should be characterized prior to testing so the reader can understand the geometry of the film. Why is it circular? Why do they all have what looks like a pushed in area in the film? How many films were tested?

2. No statistical information is given. The authors need to demonstrate that this is repeatable with multiple experiments and error bars.  

3. The Raman map in Figure should be repeated to be comparable to the image, or the image should be roatated.

4. What do the authors mean by "good control of the Fermi level" in line 181.

5. The authors refer to a "detailed AFM analysis" in the discussion, but I felt  like the AMF was fairly basic and not discussed in detail. How can you say GO interfaces are important in this case when you can't even tell where the individual flakes are? Much more detail is needed.

6. The quality of all images is poor and thus, it is impossible to make conclusions that the authors seem to draw about height of sample, involvement of aluminum in the bubbles, etc.  

7. A side-view EBIC image would be really helpful. I recommend the authors do this. 

Overall, because there are major details missing about the true nature of this sample, it is impossible to determine if the EBIC and electric measurements are meaningful.  I believe this manuscript needs a major re-write prior to publication.

Reviewer 2 Report

The main emphasis of the authors is on multiple resistive switching that were revealed by the electron beam induced current (EBIC) technique, but this work lack of novelty and significance. So, this work can’t be recommended in Nanomaterials Journal.

Ø  The AFM images are not clearly to differentiate the conduction channels.

Ø  The conduction mechanism is not properly explained in the manuscript.

Ø  In the discussion section difficult to understand the conduction filaments revealed by EBIC and conduction channels. Please explain the pristine and EBIC data clearly.

Ø  To visualize the device reliability and durability endurance and retention measurement is not discussed in the manuscript.

Ø  English must be improved.

Reviewer 3 Report

The author's work is to show that thin GO films can perform resistive memory, which can attract the readers’ interest in the journal. The authors characterized the device using EBIC and I-V measurements. However, the work needs some revisions for publication in the followings. 

1>      The schematics for the device structure and circuit model are necessary for the readers’ understanding. Especially the authors explained that the device has not only a resistive component but also a capacitive component; the configuration of the circuit components may behave differently, contrary to the author's intention.

2>      In Fig. 1 and 6, the exponential representation of the y-axis is not an appropriate representation in the scientific literature.

3>      The transition from HRS to LRS is explained as a kind of joule heating by current flow, but the transition from LRS to HRS is not clear. The authors need to do more reasoning about the transition mechanism to HRS.

4>      In Fig. 6b, more explanation is required: the definitions of delta V, Max. voltage in depletion, and accumulation capacitance and the way they were characterized (or measured).

5>      I wonder about the existence of aluminum oxide (thin and uniform) without any thermal treatment. The authors should show more detail for the fabrication of the device. In addition, in lines 233 – 249, the authors tried to claim that the Al was oxidized and eventually made the electrical short pass for LRS. However, I can accept that for a very short time period when Al is oxidized, the current flow can be measured but once the oxygen layer is formed it can impede the current flow, which will result in the opposite effect of LRS. I may guess some depletion regions of GO can be changed due to the applied bias voltage and in this case, as the capacitance may be changed by the maximum sweep voltage as shown in Fig. 6a, the capacitance may be changed by the time applying the bias voltage even with a single value.

Round 2

Reviewer 1 Report

This paper does not adequately characterize the materials used prior to experimentation. Experimental methods are not well described.  Every experiment should be repeated to provide statistical information in order to make sure that the results are representative of the system and this had no repeated experiments. There are questions about the EBIC methods, as well as characterization techniques.

Reviewer 2 Report

I would recommend publishing.

Reviewer 3 Report

The paper is revised and improved in some parts, but it still cannot fully defense my questions. So I guess the paper needs minor revisions.

 1>      Fig. 1 of revision is the schematic showing the device structure and characterization setup, but it is not a circuit model. The circuit model (it can be an equivalent circuit model) can show how capacitive and resistive components are configured (connection in series or parallel, or bulk or distributed components). It is important not only because the working mechanism can be explained simply with the proper circuit model but also because the reasonable interpretation of the measurements (capacitance or resistance) can be on the proper circuit model.

Thus, I would like to insist that the authors need to show up any simple circuit model of the device based on their best understanding.

2>      The revision does not sufficiently explain the mechanism for the transition from LRS to HRS. I can agree that the EBIC images show that the conductive channels were changed (increased or decreased depending on the bias directions). However, the EBIC image is an observation of the results of the transformation. I am (also, readers may be) curious about the author's explanation (thoughts) on the root cause (a mechanism) of these transitions.
